# TabPalooza: A Benchmark Odyssey for Tabular Model Evaluation

## Abstract

Tabular data is fundamental to machine learning, yet a lack of a widely accepted and comprehensive benchmark hinders the reliable evaluation of models, which range from tree-based models and neural network to more recent in-context learning-based approaches. Existing benchmarks are often limited in the diversity of meta-features considered, leading to inconsistent model rankings and reduced generalizability. To address these issues, this study constructs a novel benchmark for tabular data classification and regression, designed with an explicit focus on two key, often competing, characteristics: Diversity and Efficiency. We propose a pipeline to quantitatively assess benchmark diversity and introduce a method for selecting a representative subset of datasets. Our results demonstrate that the proposed benchmark achieves superior diversity compared to existing alternatives while maintaining evaluation efficiency. The main contributions include the new benchmark TabPalooza, the evaluation pipeline, and an empirical validation of the benchmark's enhanced coverage. The proposed TabPalooza is available at https://huggingface.co/datasets/data-hub-xyz987/TabPalooza.

## 1 Introduction

Tabular data, which is characterized by a structured organization of rows (representing instances) and columns (representing features), constitutes a fundamental component of machine learning. The analysis of such data remains a central task in the discipline, and the corresponding methodologies have undergone significant evolution over time.

Conventional techniques—including Logistic Regression (LR), Support Vector Machines (SVM), and decision trees—continue to be widely employed as baseline models across diverse applications (Bishop & Nasrabadi, 2006; Mohri et al., 2018). However, for an extended period, tree-based models such as Random Forests (RF)(Breiman, 2001), XGBoost (Chen & Guestrin, 2016), Light-GBM (Ke et al., 2017), and CatBoost (Prokhorenkova et al., 2018) have been recognized as highly effective methods for handling tabular data. In contrast to the established dominance of neural network (NN)-based models in domains like computer vision and natural language processing, the comparative efficacy of NN-based and tree-based models in tabular data analysis remains a subject of ongoing investigation—even with the recent introduction of numerous NN-based architectures (Arik & Pfister, 2021; Wang et al., 2021; Gorishniy et al., 2021; Somepalli et al., 2022; Chen et al., 2022; Jeffares et al., 2023; Chen et al., 2023; Yan et al., 2023)

Recently, there has been growing research momentum in in-context learning(ICL)-based models, emerging as formidable competitors, including TabPFN (Hollmann et al., 2022) and its successor TabPFN-v2 (Hollmann et al., 2025), TabICL (Qu et al., 2025), Mitra (Zhang & Danielle, 2025), and LimiX (Zhang et al., 2025).

In response to the growing demand for reliable model evaluation and accelerated iterative development, a number of high-quality benchmarks have been established (Liu et al., 2024; Bischl et al., 2017; Hollmann et al., 2025; McElfresh et al., 2023; Erickson et al., 2025; Fischer et al., 2023). Nevertheless, there remains no widely accepted consensus regarding which benchmark should be regarded as authoritative. Consequently, when models are evaluated across different benchmarks, they frequently exhibit significant rank variations, leading to inefficiencies and increased time costs in model selection and validation. This phenomenon stems from the fact that different benchmarks

are designed to evaluate distinct aspects of model performance. Moreover, most existing benchmarks are limited in scope—often comprising fewer than 50 datasets—which constrains the generalizability of conclusions drawn from them (Ye et al., 2024). Although recent efforts have begun to address the issue of dataset diversity, current evaluations remain narrow, typically considering fewer than five attributes—such as dataset size, the proportion of categorical features, the number of classes, and the class imbalance ratio (Ye et al., 2024; Erickson et al., 2025; McElfresh et al., 2023).

To address these limitations, we aim to construct a fair and commonly acceptable benchmark. It is designed with two primary characteristics: **Diversity** and **Efficiency**. These objectives are often in tension, thereby framing the central challenge as one of maximizing diversity under a predetermined volume of data.

**Diversity** is assessed along two key dimensions. The first involves evaluating datasets using a broader set of meta-features. The second dimension centers on a proposed pipeline that leverages specialized subsets derived from our benchmark to approximate the performance of other benchmarks. A lower reconstruction error indicates a higher degree of diversity in our benchmark. Our proposed benchmark exhibits a significantly lower reconstruction error, confirming its enhanced diversity.

**Efficiency** limit the total number of datasets. While collecting as many datasets as possible may enhance the diversity of a benchmark, it is necessary to constrain the number of included datasets to maintain the efficiency of tabular model evaluation. In our study, we present a pipeline for selecting the most representative datasets while preserving diversity.

Below, we summarize our main contributions:

- We propose a diverse and efficient tabular benchmark for classification and regression, leveraging an extensive collection of datasets and a rich set of meta-features.

- We propose a pipeline to assess the diversity of tabular benchmarks.

- We demonstrate that our proposed benchmark exhibits superior diversity compared to existing alternatives.

## 2 RELATED WORKS

Substantial research dedicated to the development of benchmarks for tabular data, as exemplified by OpenML-CC18 (Bischl et al., 2017), PMLB (Olson et al., 2017; Romano et al., 2022), and TabRepo (Salinas & Erickson, 2023). Recent years have witnessed dedicated efforts to establish high-quality benchmarks in this domain. For example, TabZilla (McElfresh et al., 2023) proposes a "most difficult" dataset benchmark, TabRed (Rubachev et al., 2024) emphasizes the use of real-world datasets, and TabArena (Erickson et al., 2025) incorporates timeliness of datasets. Concurrently, TALENT (Ye et al., 2024) compiles a comprehensive collection of datasets spanning diverse task types, dataset scales, and application domains. Nevertheless, research on the evaluative effectiveness of such benchmarks remains relatively limited.

Research on the relationship between dataset meta-features and model performance predominantly centers on the comparative analysis between tree-based models and neural network (NN)-based models. McElfresh et al. (McElfresh et al., 2023) systematically compared 19 models on 176 datasets, and studied which properties (e.g. feature distributions, skewness, heavy-tailedness, dataset irregularities, etc.) would make GBDT or neural networks more advantageous. Ye et al. (Ye et al., 2024) extended this analysis by comparing 32 models on more than 300 datasets and constructing a predictive mapping from dataset meta-features and training dynamics to model performance. Although primarily designed as a benchmark, TabArena (Erickson et al., 2025) also conducted large-scale comparisons between tree-based and neural models, thereby revealing the performance differences of the two approaches under a unified benchmark. However, these comparative analyses are predominantly confined to the broad dichotomy between tree-based and neural network-based models, operating at a relatively coarse level of granularity.

## 3 METHODOLOGY

### 3.1 DIVERSITY ASSESSMENT

#### 3.1.1 EVALUATION PROTOCOLS

We assess the diversity of a benchmark based on its reconstruction error with another benchmark, i.e., the mean of rank difference ($d_r$) between two benchmarks. Specifically, we define a source benchmark $\mathcal{S} = \{s_1, s_2, \ldots, s_M\}$ and a target benchmark $\mathcal{T} = \{t_1, t_2, \ldots, t_N\}$ containing $M$ and $N$ datasets respectively, along with a collection of $K$ baseline models $\mathcal{M} = \{g_1, g_2, \ldots, g_K\}$. The performance of the model $g_k$ is evaluated on the dataset $s_m$ and the dataset $t_n$, yielding a performance ranking $r_{m,k}^S$ and $r_{n,k}^T$. We then estimate the mean rank of model $g_k$ on $\mathcal{T}$ by using the information of $\mathcal{S}$, denoted as $\bar{p}_k = \frac{1}{N} \sum_{n=1}^{N} p_k^n$, where $p_k^n$ is the rank estimation of model $g_k$ on dataset $t_k$. The final reconstruction error is computed as the difference between the real mean rank values obtained over all datasets in $\mathcal{T}$ and the predict mean rank values, respectively.

$$d_r = \frac{1}{K} \sum_{k=1}^{K} \left| \bar{p}_k - \bar{r}_k^T \right|.$$

where $\bar{r}_k^T = \frac{1}{N} \sum_{n=1}^{N} r_{n,k}^T$. A smaller value of $d_r$ indicates that the source benchmark $\mathcal{S}$ better preserves the ranking behavior of the target benchmark $\mathcal{T}$.

#### 3.1.2 BASELINE MODELS AND EVALUATION METRICS

**Baseline models** To evaluate the capabilities of the tabular benchmarks, we assess 11 baseline models comprising: (1) four ICL-based approaches (with three ICL models for regression tasks, as TabICL is exclusively designed for classification), (2) three tree-based approaches, (3) three neural network-based models, and (4) one ensemble automated machine learning (Auto-ML framework) approach. Although numerous tree-based and neural network-based models exist, we select three representatives from each category with diverse ranking performances across different benchmarks, as evaluated in Zhang et al. (2025).

- **Tree-based approaches.** We include XGBoost (Chen & Guestrin, 2016), CatBoost (Dorogush et al., 2018), Random Forest (RF) (Breiman, 2001). All models undergo optimization through the Optuna framework (Akiba et al., 2019) employing 5-fold stratified cross-validation, utilizing identical hyperparameter search spaces as those specified in Zhang et al. (2025).

- **NN-based approaches.** We evaluate ExcelFormer (Chen et al., 2023), MLP (Goodfellow et al., 2016; Gorishniy et al., 2021), ResNet (He et al., 2016; Gorishniy et al., 2021), These NN-based models are trained and evaluated using the TALENT (Liu et al., 2024) Toolbox.

- **ICL-based models.** We include LimiX (Zhang et al., 2025), TabPFN-v2 (Hollmann et al., 2025), TabICL (Qu et al., 2025), and Mitra (Zhang & Danielle, 2025).

- **Auto-ML framework.** Additionally, we include AutoGluon-Tabular (Erickson et al., 2020), an automated framework that streamlines model search and ensemble construction workflows. For each dataset, we employ the default hyperparameter search space while implementing a standardized 600-second computational constraint for the optimization process.

**model performance** For performance assessment, we utilize ROC AUC (area under the receiver operating characteristic curve), accuracy (ACC), and F1 score as classification metrics. In multi-class scenarios, the One-vs-One strategy is implemented for both ROC AUC and F1 score computations. The ROC AUC reflects model performance across varying decision thresholds, whereas ACC and F1 score exhibit responsiveness to class imbalance. To evaluate regression performance properly, we employ normalized RMSE and $R^2$ as the two evaluation metrics. We also calculate the ranks of models with respect to these metrics.

Table 1: The types of extracted meta-features and their corresponding quantities.

| Meta-feature type | Number |
|---|---|
| General | 13 |
| Statistical | 48 |
| Info-theory | 13 |
| Landmarking | 14 |
| Model-based | 24 |

### 3.1.3 META-FEATURE EXTRACTION AND SELECTION

Similar to TabZilla (McElfresh et al., 2023), we extract meta-features using the Python library PyMFE (Alcobaça et al., 2020), comprising 111 distinct meta-features. We additionally compute the cell missing ratio for each dataset. As PyMFE is designed for classification tasks, we adapt its application to regression datasets by discretizing the continuous target variables into 10 equal-frequency bins when calculating their meta-features. The types of extracted 112 meta-features and their corresponding quantities are presented in Table 1, with our defined cell missing rate included in the 'General' category.

To identify the most informative meta-features, we evaluate their correlations with model performance rankings using multiple correlation estimation methods:

- **Pearson correlation:** quantifies linear relationships between variables, representing the most widely adopted and interpretable correlation metric;

- **Spearman correlation:** detects monotonic associations through rank-based statistics, demonstrating robustness to non-linear yet monotonic patterns;

- **Kendall correlation:** offers a more rigorous assessment of rank concordance while exhibiting reduced sensitivity to outlier effects;

- **Distance correlation (dCor):** effectively captures general non-linear dependencies, serving as a valuable complement to conventional correlation measures.

### 3.1.4 META-FEATURE EVALUATION

In our study, we aim to demonstrate the correlation between the meta-features of each dataset and the performance rank of different baseline models. This supervised study consists of $N$ dataset samples, each characterized by $z$ meta-features, and evaluate $\mathcal{M}$ baseline models on these datasets. For the $n$-th dataset and $m$-th model, we construct the instance $\mathbf{x}_i = [f_i \quad s_i] \in \mathbb{R}^{z+1}$, where $f$ is the extracted meta-features and $s$ is the index of model. For each instance, we can obtain the corresponding rank $r_i$. Given a benchmark $\mathcal{B} = \{(\mathbf{x}_i, r_i) : i = 1, \ldots, N \times M\}$, our goal is to learn a regression model $f : \mathbb{R}^{z+1} \to \mathbb{R}$ on $\mathcal{B}$ by minimizing the empirical risk, mapping the pair of dataset feature and model to ranks.

$$\min_f \sum_{(\mathbf{x}_i, r_i) \in \mathcal{B}} \mathcal{L}(r_i, f(x_i))$$

### 3.1.5 BENCHMARK ALIGNMENT

We propose a benchmark alignment pipeline to estimate the model performance on the target benchmark. For each dataset within a target benchmark, we compute the Euclidean distance to every dataset in a source benchmark, based on a selected set of meta-features. The dataset from the source benchmark with the smallest distance is then assigned to a newly constructed subset. This procedure is repeated until the subset contains the same number of datasets as the target benchmark. Finally, the ranks of baseline models evaluated on this subset are used as an estimate of their ranks on the target benchmark, enabling the calculation of the mean rank difference between the two benchmarks.

## 3.2 EFFICIENCY GUARANTEE

The TabPalooza benchmark is constructed using a methodology designed to minimize internal dataset similarity. Specifically, we perform hierarchical clustering on the available datasets, controlling the number of clusters as a parameter. From each resulting cluster, one dataset is randomly selected for inclusion in TabPalooza. Concurrently, we ensure that the mean rank difference remains within a predefined acceptable range, thereby preserving the benchmark's evaluative reliability.

## 4 EXPERIMENTS

deep tabular prediction, machine learning

### 4.1 DATASET CURATION

**Datasets collection** This study utilizes datasets from resultsestablished benchmarks for both classification and regression tasks. The classification benchmarks include TALENT-CLS, OpenML-CC18, PFN-CLS, TabZilla, and TabArena, while the regression benchmarks comprise TALENT-REG, PFN-REG, and CTR23. To expand the diversity of the data, additional 2273 classification and 1381 regression publicly available datasets were sourced from Kaggle. For any dataset lacking a predefined testing split, a standardized train-test split ratio of 7:3 is applied. In the case of classification tasks, this split is stratified to preserve the proportional distribution of classes across both subsets.

**Kaggle datasets validation** The datasets obtained from Kaggle exhibit diverse formats and require preprocessing prior to utilization. To facilitate the identification of dataset files and their corresponding targets, we employ the DeepSeek-r1:32b (Guo et al., 2025) and Qwen3:32b (Yang et al., 2025) models. These large language models (LLMs) are accessed via the Ollama API. The prompts provided to the LLMs comprise the dataset files, the initial ten lines of each file, and an output sample. The LLMs are tasked with determining the paths to the training and testing files, as well as specifying the classification or regression target. To enhance recognition accuracy, only the consistent results between DeepSeek-r1 and Qwen3 are retained.

To further assess the solvability of these datasets, we train and evaluate the XGBoost model (Chen & Guestrin, 2016) across all datasets. For classification tasks, datasets exhibiting an ROC AUC (area under the receiver operating characteristic curve) below 0.55 are excluded from further consideration, whereas for regression tasks, those with an $R^2$ coefficient under 0.2 are similarly eliminated.

The final curated dataset collection comprises 335 classification datasets and 251 regression datasets.

**Dataset selection and deduplication** To manage computational demands, datasets within existing benchmarks were subjected to specific exclusion criteria. Those containing over 50,000 training samples or exceeding 10,000 features were omitted. For classification tasks, datasets with more than 10 categories were also excluded to meets the requirements of the ICL-based baseline models.

Following this selection protocol, the classification task subset comprises 179 datasets from TALENT-CLS, 62 from OpenML-CC18, 29 from PFN-CLS, 27 from TabZilla, and 33 from TabArena. Similarly, for regression tasks, the curated collection includes 33 datasets from CTR23, 28 from PFN-REG, and 99 from TALENT-REG.

Finally, we eliminate duplicate datasets across all benchmarks by matching both dataset names and dataset sizes. Distinct versions of the same dataset are retained to preserve their unique meta-feature variations. In our study, we analyze a total of 501 classification datasets and 335 regression datasets, collectively designated as the **U**niversal **D**ataset **P**ool (**UDP**).

### 4.2 META-FEATURE EVALUATION

For meta-feature evaluation, we establish a baseline by setting the predicted mean rank on the test set to be equal to the empirical mean rank observed in the training set. We then calculate the $d_r$ between these predicted values and the actual observed values. For the meta-feature based rank prediction method, we generate individual rank predictions for each dataset within the testing set, followed

Table 2: The $d_r$ of rank predictor and baseline in classification task. The smaller $d_r$ is better.

| | method | AUC | ACC | F1 |
|---|---|---|---|---|
| TabZilla | baseline | 0.582 | 0.637 | 0.524 |
| | predictor | 0.539 | 0.484 | 0.387 |
| | **delta** | **0.043** | **0.153** | **0.137** |
| TabArena | baseline | 0.664 | 0.613 | 0.382 |
| | predictor | 0.566 | 0.653 | 0.546 |
| | **delta** | **0.098** | **-0.040** | **-0.164** |
| OpenML-CC18 | baseline | 0.686 | 0.805 | 0.612 |
| | predictor | 0.630 | 0.563 | 0.409 |
| | **delta** | **0.056** | **0.242** | **0.203** |
| PFN-CLS | baseline | 0.502 | 0.590 | 0.476 |
| | predictor | 0.334 | 0.440 | 0.369 |
| | **delta** | **0.168** | **0.150** | **0.107** |
| TALENT-CLS | baseline | 0.528 | 0.701 | 0.579 |
| | predictor | 0.494 | 0.652 | 0.462 |
| | **delta** | **0.034** | **0.049** | **0.114** |
| Average | baseline | 0.592 | 0.677 | 0.515 |
| | predictor | 0.513 | 0.558 | 0.435 |
| | **delta** | **0.079** | **0.119** | **0.080** |

Table 3: The $d_r$ of rank predictor and baseline in regression task. The smaller $d_r$ is better.

| | method | $R^2$ | RMSE |
|---|---|---|---|
| CTR23 | baseline | 0.404 | 0.387 |
| | meta-pred | 0.366 | 0.357 |
| | **delta** | **0.038** | **0.030** |
| PFN-REG | baseline | 0.344 | 0.392 |
| | meta-pred | 0.320 | 0.349 |
| | **delta** | **0.024** | **0.043** |
| TALENT-REG | baseline | 0.602 | 0.576 |
| | meta-pred | 0.366 | 0.471 |
| | **delta** | **0.236** | **0.105** |
| Average | baseline | 0.450 | 0.452 |
| | meta-pred | 0.395 | 0.392 |
| | **delta** | **0.055** | **0.060** |

by the calculation of corresponding $d_r$. The evaluation results for each benchmark are presented in Tables 2 and 3.

For classification tasks, the incorporation of meta-features leads to a significant improvement in rank estimation across most benchmarks, although exceptions are observed for the ACC and F1 score metrics within the TabArena benchmark. Specifically, the average performance gains correspond to 0.079 in AUC, 0.119 in ACC, and 0.090 in F1 score, respectively.

In regression tasks, the inclusion of meta-features also demonstrates a substantial improvement in rank estimation across all evaluated benchmarks. The observed average performance gains are 0.055 for $R^2$ and 0.060 for RMSE, respectively

### 4.3 BENCHMARK DATASET

Firstly, we determine the appropriate size of the Benchmark TabPalooza. To this end, we systematically vary the number of datasets included in the TabPalooza and compute the corresponding `diff_rank` values under all correlation methods and threshold settings. By plotting the average `diff_rank` against the size, we select the inflection point where the curve becomes relatively flat as the optimal size of TabPalooza. This ensures that the resulting benchmark is compact while still maintaining strong alignment with the full benchmark. Based on this analysis, we set the size of TabPalooza to 100 datasets for classification tasks and 140 datasets for regression tasks.

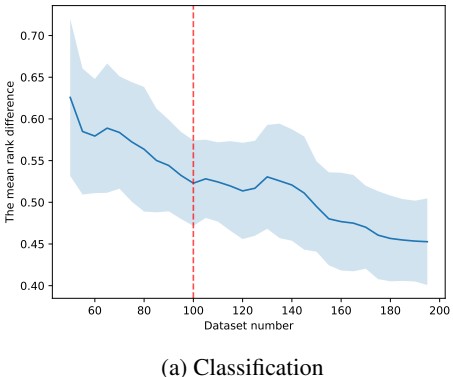
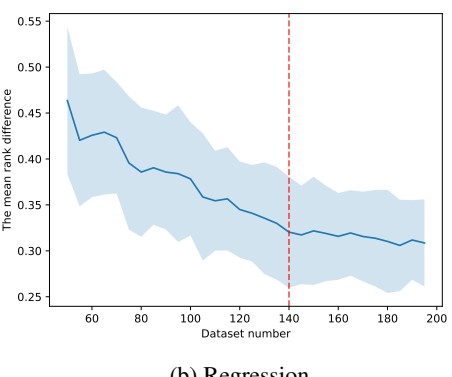

(a) Classification                    (b) Regression

Figure 1: Size selection of TabPalooza for (a) classification and (b) regression.

### 4.4 BENCHMARK ALIGNMENT EVALUATION

With the sizes determined in Section 4.1, we now construct TabPalooza under all configurations of correlation methods and thresholds. For each configuration, important meta-features are first identified, and hierarchical clustering is then applied on the **UDP** to obtain the desired number of representative datasets (100 for classification and 140 for regression). The resulting TabPalooza is compared against each target benchmark using the `diff_rank` metric.

This evaluation directly measures the alignment between the constructed TabPalooza and the full benchmarks. By aggregating `diff_rank` values across all benchmarks and metrics, we can assess how well the TabPalooza captures the ranking patterns of the complete benchmark suite. A lower `diff_rank` indicates stronger alignment, demonstrating that TabPalooza effectively preserves benchmark characteristics while remaining compact.

For classification tasks, we construct the TabPalooza using the setting `Kendall+0.07`. We then evaluate the alignment of the resulting TabPalooza with each benchmark under three metrics (`AUC`, `ACC`, and `F1`). The results are summarized in Table 4.

Table 4: Diff_rank values of TabPalooza for classification across benchmarks and metrics.

| Benchmark | AUC | ACC | F1 |
|---|---|---|---|
| TabZilla | 0.517 | 0.467 | 0.443 |
| TabArena | 0.389 | 0.507 | 0.324 |
| CC18 | 0.404 | 0.452 | 0.421 |
| PFN-CLS | 0.477 | 0.533 | 0.425 |
| TALENT-CLS | 0.289 | 0.499 | 0.286 |
| Average | 0.415 | 0.492 | 0.380 |

For regression tasks, we construct TabPalooza using the configuration `dCor+0.12`, which was identified as the best-performing setting in Section 4.1. We evaluate the alignment of TabPalooza with three regression benchmarks (`CTR23`, `PFN-REG`, and `Talent-REG`) under two metrics (`R2` and `RMSE`). The results are summarized in Table 5.

Table 5: Diff_rank values of TabPalooza for regression across benchmarks and metrics.

| Benchmark | R² | RMSE |
|-----------|------|------|
| CTR23 | 0.136 | 0.152 |
| PFN-REG | 0.273 | 0.249 |
| TALENT-REG | 0.152 | 0.166 |
| Average | 0.187 | 0.189 |

## 4.5 COMPARE WITH OTHER BENCHMARKS

We further evaluate the alignment capability of additional benchmarks. Figure 2 and Figure 3 present the $d_r$ values for benchmark pairs. Notably, when employing TabPalooza as the alignment source for other benchmarks, the resulting $d_r$ demonstrates consistently lower values. Conversely, when TabPalooza serves as the target for alignment by other benchmarks, the observed $d_r$ exhibits comparatively higher values.

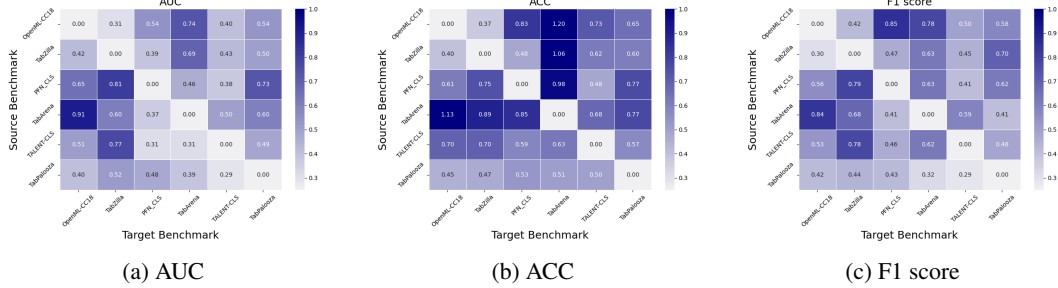

(a) AUC  (b) ACC  (c) F1 score

Figure 2: The cross-reconstruction error of the classification benchmark is visualized using a color scale, in which lighter or lower-intensity colors correspond to superior reconstruction performance.

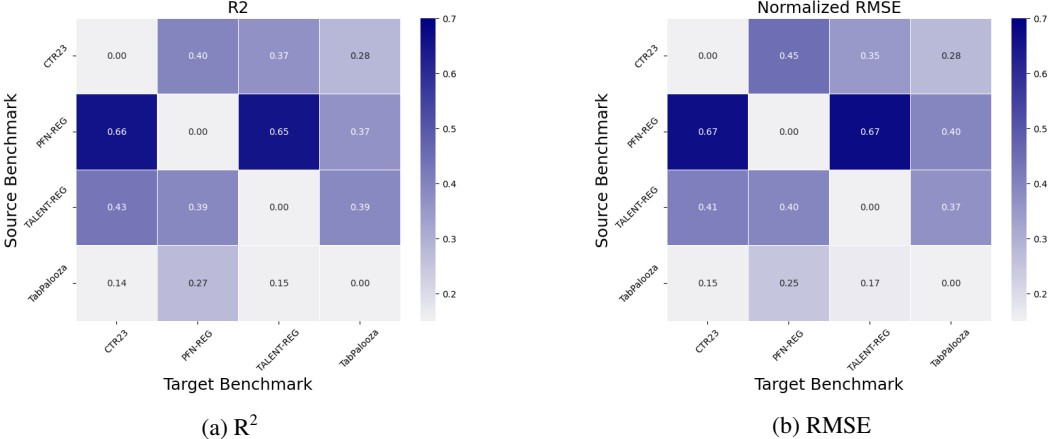

(a) R$^2$  (b) RMSE

Figure 3: The cross-reconstruction error of the regression benchmark is visualized using a color scale, in which lighter or lower-intensity colors correspond to superior reconstruction performance.

## 5 CONCLUTION

This study addresses the critical need for a robust and representative benchmark to evaluate models for tabular data. The proliferation of diverse modeling approaches—from traditional tree-based methods and NN-based model to modernand in-context learning models—has highlighted the limitations of existing benchmarks, which are often constrained in scale and meta-feature diversity, resulting in inconsistent and non-generalizable evaluations.

To overcome these challenges, we propose a novel benchmark explicitly designed around the dual principles of diversity and efficiency. We introduce a quantitative pipeline to assess diversity, demonstrating that our benchmark achieves improved coverage and reconstruction compared to existing alternatives. Furthermore, we develop a selection methodology that maintains this high level of diversity within a limited set of datasets, thereby ensuring practical evaluation efficiency.

Our contributions establish a more reliable foundation for comparative model analysis, and we anticipate that this work will promote fairer comparisons and accelerate iterative development in tabular data research. Future work will focus on expanding the benchmark with additional datasets and meta-features, as well as exploring its applications in automated model selection and combination.

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
