# OpenReview forum: "TabPalooza: A Benchmark Odyssey for Tabular Model Evaluation"
_ICLR.cc/2026/Conference — ICLR 2026 Conference Withdrawn Submission_

### Official Review · Reviewer_Vqe5 · 2025-10-30

**Soundness:** 1
**Presentation:** 1
**Contribution:** 1
**Rating:** 0
**Confidence:** 5

**Summary:**

The paper presents TabPalooza, a proposed benchmark for tabular machine learning that aims to combine diversity and efficiency through meta-feature analysis, clustering, and a rank-based alignment objective.

**Strengths:**

•	Designing more efficiently executable yet diverse benchmarks is a relevant goal for the tabular data field.

**Weaknesses:**

•	The paper is clearly work in progress and not ready for being reviewed at ICLR.
•	The paper is not rich in content: it does not use the full allowed length, contains blank lines, and has no Appendix, although several aspects require more clarification.
•	The spelling and overall design can be improved
•	Core claims are incorrect: the authors state that “existing benchmarks are often limited in the diversity of meta-features considered.” However, none of the meta-features utilized here are more diverse than in prior studies — for instance, McElfresh et al. used nearly all features in PyMFE, whereas this study only employs 112 despite claiming broader coverage.
•	There is no novelty. The proposed methodology does not differ meaningfully from McElfresh et al., who also correlated meta-features with performance. Adding missing-value counts and different correlation strategies is not sufficient.
•	Very limited information on hyperparameter optimization
•	The approach of correlating meta-features with performance is fragile and requires each model to be extensively tuned, ideally using cross-validation ensembles, as shown in Tschalzev et al. (2025). An approach like this requires rigorous statistical testing and a very carefully designed modeling strategy for each model. Since that was not done here, it is likely that the reported results would not hold under more rigorous evaluation.

Recommendation:
This paper clearly is in an unfinished state and should not be accepted to ICLR.

McElfresh, D., Khandagale, S., Valverde, J., Prasad C, V., Ramakrishnan, G., Goldblum, M., & White, C. (2023). When do neural nets outperform boosted trees on tabular data?. Advances in Neural Information Processing Systems, 36, 76336-76369.
Tschalzev, A., Purucker, L., Lüdtke, S., Hutter, F., Bartelt, C., & Stuckenschmidt, H. (2025). Unreflected use of tabular data repositories can undermine research quality. arXiv preprint arXiv:2503.09159.

Tschalzev, A., Purucker, L., Lüdtke, S., Hutter, F., Bartelt, C., & Stuckenschmidt, H. (2025). Unreflected use of tabular data repositories can undermine research quality. arXiv preprint arXiv:2503.09159.

**Questions:**

-

---

### Official Review · Reviewer_LjjZ · 2025-10-30

**Soundness:** 2
**Presentation:** 1
**Contribution:** 2
**Rating:** 0
**Confidence:** 4

**Summary:**

This paper discusses the problem of many different benchmarks in tabular ML. The authors argue that any single benchmark currently available to the academic community lacks diversity, while evaluating on all the benchmarks is very inefficient. The authors, therefore, frame the problem of building a new benchmark as the one that maximizes diversity, while maintaining maximal efficiency, reflected by the total number of datasets in the resulting benchmark.

The main unanswered question is, therefore, how to come up with a proxy metric that would describe the diversity of a benchmark. The authors first collect 112 diverse meta-features, describing various property of each dataset. Then, they filter these meta-features, to only leave the ones that have the highest relation to the ranking of 11 baseline methods. Then, authors propose a procedure which produces an estimate of ranking of 11 methods on a target benchmark, based on their performances on a source benchmark. To do this, authors select a subset of datasets from the source benchmark, which has the closest distance in meta-feature space to the target benchmark. They then use this subset to estimate average ranks of the baseline models on the target benchmark. The authors use MAE errors in estimating ranks to evaluate how good is the source benchmark at reflecting the diversity contained in the target benchmark. The authors use averaged difference in ranks across many established target benchmarks as a proxy for the diversity of a source benchmark.

The paper is not very clearly written, with many minor mistakes, which are described in the Weaknesses section. The main motivation behind my low score, however, comes from the disagreement with the idea of the proposed benchmark. The authors of the TabArena benchmark, for example, write “Most importantly, many datasets used in benchmarks are outdated, contain problematic licenses, do not represent real tabular data tasks, or are biased through data leaks”. The main motivation behind that benchmark, it seems, is to build a collection of datasets which are high in quality by the definition provided by the authors. In this paper, however, authors collect thousands of datasets from Kaggle, and use an automated LLM-based procedure to filter them. Therefore, the proposed benchmark takes a fundamentally opposite approach to the one of authors of TabArena, and, therefore, cannot be a replacement for it. Many other benchmarks also have some philosophy behind their filtration procedure, and cannot be replaced by a unified benchmark that lacks it, or, to some degree, takes an opposite approach. I will elaborate further in the weaknesses section.

This paper, however, has many positive contributions. The idea of estimating performance on one benchmark based on the performance on the source dataset is great, and can be useful in development to estimate performance on a variety of benchmarks quicker. I do, however, believe that any final evaluation should include full versions of the original benchmarks, since differences in ranking across different benchmarks have interpretable meaning, and are useful for researchers.

**Strengths:**

1. The idea of estimating performance on one benchmark by performance on another is novel, interesting, and potentially very useful.
2. This paper takes a closer look at the relationship between meta-features and relative performance of different methods. A deeper study on this topic can be very useful to highlight relative strengths and weaknesses between Tabular ML methods.
3. Figures 2 and 3 tell a story about relative similarity of different benchmarks.

**Weaknesses:**

The main weakness of this paper, for me, is that many current benchmarks represent a conflict of ideological approaches towards how a benchmark for Tabular ML should look. TabZilla takes the approach that only the hardest datasets need to be included, TabArena believes in the strict filtration and curation, TabReD believes that datasets need to be maximally realistic. I disagree that there is a need for one benchmark that fits all, instead, I believe we need a set of benchmarks, with a single benchmark corresponding to each idea. I also believe that the procedure described by the authors does not prevent datasets with data leakage from being included in the TabPalooza benchmark, and even encourages it. By maximizing diversity, we will include many datasets with problems in them, since ranking of baselines might change greatly on these datasets. I believe acceptance of the paper in its current form could lead to scientists optimizing their methods to abuse leaks or work better on some problematic datasets. In other words, I do believe that some minimal dataset filtering procedure performed by a human is necessary for any given benchmark, while this benchmark simply takes all available data in the tabular form and makes it into a dataset.

There are also several minor issues with writing, which did not greatly influence my score, but I would nevertheless like to point out some of them to help increase the quality of the paper.
1. On lines 120-121, the notation is unclear, what does n correspond to? Section 3.1.1 is also not very clear as a whole.
2. On line 156, the word “model” should be spelled with a capital letter.
3. On line 178, what does equal frequency mean?
4. On line 198, “evaluate” should be “we evaluate” or “evaluates”
5. On line 199, should there be a comma between f and s?
5. Subsection 3.1.5 was not clear to me.
6. On line 230, the word “resultsestablished” should be corrected.
7. On line 328, is diff_rank the same as d_r from before?
8. On line 353, I did not find the determination of sizes in section 4.1
9. On line 226, there seems to be a placeholder text left.
10. On line 364, what is Kendall+0.07? I understand that Kendall refers to the Kendall Correlation from 3.1.3, but what does 0.07 mean?
11. On line 379, I do not believe it was identified as such, or I might have missed this part in section 4.1.

**Questions:**

1. How do you envision the role and place of this benchmark?
2. Could you elaborate on the nature of datasets you referred to on lines 233-234? Does this only includes tabular data or all data?
3. On line 317, could this effect be related to the fact that TabArena has the toughest filtration standards? If so, could this mean that other (positive) results only show that meta-features can determine leakage or non-tabular data?
4. Is diff_rank the same as d_r?
5. Do you believe that limiting training size by 50,000 samples changes the ranking of the baseline methods? I understand your concerns about efficiency, however, with the speed provided by modern GPUs, many practical applications use much more data. I would be interested in an ablation study on the topic of how difference in dataset sizes changes relative ranking of the baselines.

---

### Official Review · Reviewer_SFZ4 · 2025-10-31

**Soundness:** 1
**Presentation:** 1
**Contribution:** 1
**Rating:** 2
**Confidence:** 5

**Summary:**

This paper proposes TabPalooza, a tabular benchmark designed around "diversity" and "efficiency"; it develops pipelines for dataset selection, verifies that TabPalooza shows evaluation efficiency.

**Strengths:**

1. The paper is clear and easy to understand.

2. The motivation is relatively sufficient: evaluating existing tabular models is indeed costly, and it is meaningful to develop a widely recognized and efficient evaluation method.

**Weaknesses:**

1. As a benchmark-focused work, the paper feels hasty and fails to present essential results (e.g., performance of various methods across metrics and evaluation criteria).
2. It lacks dataset analysis, such as the distribution of meta-features and how it compares to other benchmarks.
3. Regarding novelty: the paper claims its core contribution is finding a representative, efficient evaluation subset, but TALENT has done similar work on evaluation subsets.
4. For dataset selection: while benchmarks like TabRed and TabArena emphasize dataset quality (to avoid issues like leakage or shift), the authors do not provide a clear dataset list, making it hard to judge if their benchmark has such problems.
5. On method selection: it is unclear why MLP and ResNet were chosen for deep learning methods (over stronger alternatives like RealMLP, TabR, ModernNCA, TabM), and FT-Transformer should be more representative than Excelformer.
6. Experimental evaluation criteria are vague and lack details—critically, there is no explanation of early stopping for deep learning methods, nor validation set splitting or cross-validation (unlike tree models which used 5-fold cross-validation), leading to unfair comparison between deep and tree-based methods.
7. Excluding datasets with over 50,000 training samples or 10+ classes clearly favors existing PFN-style models, introducing bias into the evaluation.

**Questions:**

Refer to the Weaknesses.

---

### Official Review · Reviewer_xEGe · 2025-10-31

**Soundness:** 2
**Presentation:** 2
**Contribution:** 2
**Rating:** 2
**Confidence:** 4

**Summary:**

This paper aims to address the lack of a widely accepted, diverse, and unified benchmark for tabular data. Existing benchmarks often have limited datasets and incomplete meta-features, resulting in significant discrepancies in model rankings and poor generalizability. The authors aim to construct a new benchmark, TabPalooza, that balances diversity and efficiency. This benchmark aims to maintain good model discrimination capabilities despite limited dataset size and provide a stable comparison platform for subsequent research.

**Strengths:**

S1. Transforming "diversity" from a general description into a quantifiable metric (DR), and incorporating meta-feature correlation analysis to construct a dataset selection strategy, this approach has the potential to influence future benchmark design.

S2. The data collection, cleaning, and solvability verification (including automated parsing of Kaggle datasets using LLM) are meticulous, covering multiple models (Tree-based, NN-based, ICL-based) and various evaluation metrics.

S3. The dataset is open source with clear source information, enhancing community reproducibility.

**Weaknesses:**

W1. Although the paper aligns with multiple existing benchmarks, it does not present the performance distribution of various models (Tree, NN, ICL) on TabPalooza, making it impossible to verify the effectiveness of the benchmark in terms of actual model discriminative ability. Supplement the performance ranking of different model families on TabPalooza to compare whether they are more stable or more consistent with the expected discriminative ability. This is crucial for the persuasiveness of the benchmark.

W2. TabPalooza's performance on specific domains (such as extremely high-dimensional scientific data) or long-tailed data has not been evaluated. Suggestion: Add more boundary condition experiments, such as selecting a domain-specific tabular dataset to compare changes in alignment metrics. Supplement comparison and discussion with cross-domain and multimodal benchmarks; analyze TabPalooza's potential adaptability to these new task types.

W3. Although the dataset is open, there is insufficient information on the pipeline code, computational configuration, and hyperparameter settings, which may hinder reproducibility.

**Questions:**

Q1. Why don't you provide a performance comparison of various models on TabPalooza? Is this due to data size or computational resource limitations, or is the method focused on alignment validation?

Q2. Is the meta-feature selection process based on a single relevance metric or a combination of multiple? How different are the feature sets obtained using different metrics?

Q3. Have you verified the stability of TabPalooza's performance when transferring across tasks (e.g., from classification to regression, or few-shot table prediction)?

---

### Official Review · Reviewer_61BG · 2025-11-01

**Soundness:** 2
**Presentation:** 2
**Contribution:** 2
**Rating:** 2
**Confidence:** 4

**Summary:**

To address the lack of diversity in meta-features of existing tabular-data benchmarks, this paper proposes a new classification and regression benchmark called TabPalooza. The core methodology is a reconstruction-error–based benchmark diversity evaluation pipeline that uses the mean difference in model rankings, along with efficient dataset subset selection via hierarchical clustering. In experiments, TabPalooza consistently shows lower reconstruction error than existing benchmarks and claims to achieve efficient coverage with 100 classification and 140 regression datasets.

**Strengths:**

Introduction of a quantitative diversity metric: To quantitatively evaluate benchmark diversity, the paper proposes a reconstruction-error metric using the mean difference in model rankings, and demonstrates that TabPalooza consistently achieves lower than existing benchmarks, thereby proving superior rank preservation of the benchmark.

Extensive use of meta-features and models: For dataset characterization, the study employs 112 meta-features—far broader than prior work—and attempts a comprehensive validation of the benchmark by including 11 diverse, state-of-the-art models as evaluation backbones, such as ICL-based (TabPFN-v2, LimiX), tree-based (XGBoost, CatBoost), and NN-based (ExcelFormer, ResNet).

Systematic benchmark construction effort: Building on a large dataset pool obtained from sources such as OpenML, TabZilla, and Kaggle, the paper sets an efficiency target (100 classification and 140 regression datasets) and presents a systematic dataset selection methodology using hierarchical clustering to ensure representativeness.

**Weaknesses:**

Novelty
The proposed idea amounts to a minor variation or combination of existing benchmark-construction studies based on meta-feature analysis (e.g., TabZilla, TALENT). While introducing a new meta-evaluation metric—rank reconstruction error—is creative, it does not fundamentally change the benchmark framework; rather, it is a clever refinement within the existing meta-learning perspective, so the distinctiveness is unclear and non-essential.

Technical Quality

Compromised reproducibility (critical flaw): For the meta-feature–based rank prediction model presented in Section 3.1.4, there is no concrete description of which algorithm was used (e.g., MLP, tree-based model), nor of key hyperparameters or the training procedure, severely undermining the study’s reproducibility.

Lack of robustness in meta-feature prediction: In the TabArena benchmark, for classification tasks, the meta-feature–based predictor for ACC and F1 performs worse than the baseline. This suggests that the proposed meta-feature set and rank-prediction modeling fail to capture general benchmark ranking patterns, raising concerns about the technical soundness of the core methodology.

Logical leap in dataset selection: For efficiency, after hierarchical clustering, “one dataset was randomly selected from each cluster” (Section 3.2), which does not logically support the claim that the chosen dataset is the cluster’s “most representative” (e.g., cluster medoid).

Significance
There is insufficient analysis to show that the reported improvements in for TabPalooza (Figures 2 and 3) are large enough to materially advance research in the field or provide practical significance for other researchers. The numerical gains in 𝑑𝑟 (AUC improvements of roughly ≤ 0.1) are not persuasively argued to translate into tangible improvements in ranking stability or predictive accuracy that a new model developer would experience.

Writing & Presentation
The omission of detailed explanations for the core rank-prediction model 𝑓(⋅)(Section 3.1.4) is the biggest obstacle preventing readers from fully understanding the paper. In addition, the color scales in Figures 2 and 3 are not intuitive to interpret, and the lack of clear explanations for benchmark abbreviations (e.g., CC18) reduces clarity.

**Questions:**

Detailed specification of the rank prediction model f(⋅): Please provide the concrete architecture, loss function, training protocol, and key hyperparameters of the meta-feature–based rank prediction model presented in Section 3.1.4. We need this information to assess the technical completeness and reproducibility of the work.


Resolving the TabArena inconsistency: In Table 2, the drd_rdr​ predictor’s performance on the TabArena benchmark for ACC and F1 scores is worse than the baseline (ACC: -0.040, F1: -0.164). Please analyze the cause of this degradation and propose analyses or methodological improvements to address it. This will be a decisive criterion for evaluating the robustness of the proposed meta-feature–based diversity evaluation methodology.


Justifying dataset selection: In Section 3.2, please explain the rationale for using random selection after clustering. Alternatively, select the cluster medoid instead of a random sample, or demonstrate the statistical significance (including standard errors) of the mean drd_rdr​ obtained by repeating the random selection, to establish the consistency and statistical validity of the dataset selection process.


Quantifying the efficiency/diversity trade-off: Please provide a quantitative definition of the “inflection point” used to determine the optimal size (100/140) in Figure 1, and present a clear trade-off curve between diversity (lower dr) and efficiency (number of datasets) compared with existing benchmarks. This will allow us to objectively evaluate the choice of benchmark size.

---

### Note · Authors · 2025-11-21

**Comment:**

We would like to withdraw our submission from ICLR. After further internal review, we identified issues in the current version of the manuscript that require substantial revision before it can be properly evaluated. To avoid placing unnecessary burden on the reviewers and to ensure that the work is presented in its best form, we have decided to withdraw the paper at this time.
We appreciate the opportunity to submit to ICLR and thank the program committee for their consideration. We plan to revise the manuscript thoroughly and resubmit in a future cycle.

**Withdrawal Confirmation:**

I have read and agree with the venue's withdrawal policy on behalf of myself and my co-authors.